# Hypoxia Inducible Factor 1A Supports a Pro-Fibrotic Phenotype Loop in Idiopathic Pulmonary Fibrosis

**DOI:** 10.3390/ijms22073331

**Published:** 2021-03-24

**Authors:** Gali Epstein Shochet, Becky Bardenstein-Wald, Mary McElroy, Andrew Kukuy, Mark Surber, Evgeny Edelstein, Barak Pertzov, Mordechai Reuven Kramer, David Shitrit

**Affiliations:** 1Pulmonary Department, Meir Medical Center, Kfar Saba 4428164, Israel; davids3@clalit.org.il; 2Sackler Faculty of Medicine, Tel Aviv University, Tel Aviv 69978, Israel; bekibarden@gmail.com (B.B.-W.); pertzovb@gmail.com (B.P.); Kremerm@clalit.org.il (M.R.K.); 3Respiratory Pharmacology, Charles River Laboratories, Edinburgh EH33 2NE, UK; Mary.McElroy@crl.com; 4The Leviev Heart Institute, Sheba Medical Center, Ramat Gan 5262000, Israel; dr.a.kukuy@gmail.com; 5Avalyn Pharma, 701 Pike Street suite 1500, Seattle, WA 98101, USA; msurber@avalynpharma.com; 6Pathology Department, Meir Medical Center, Kfar Saba 4428164, Israel; edelshte@clalit.org.il; 7Pulmonary Division, Rabin Medical Center, Petach Tikva 4941492, Israel

**Keywords:** IPF, HIF1A, PAI-1, extracellular matrix, bleomycin, nintedanib, in-vivo, signaling, fibrosis

## Abstract

Idiopathic pulmonary fibrosis (IPF) is a progressive lung disease with poor prognosis. The IPF-conditioned matrix (IPF-CM) system enables the study of matrix–fibroblast interplay. While effective at slowing fibrosis, nintedanib has limitations and the mechanism is not fully elucidated. In the current work, we explored the underlying signaling pathways and characterized nintedanib involvement in the IPF-CM fibrotic process. Results were validated using IPF patient samples and bleomycin-treated animals with/without oral and inhaled nintedanib. IPF-derived primary human lung fibroblasts (HLFs) were cultured on Matrigel and then cleared using NH_4_OH, creating the IPF-CM. Normal HLF-CM served as control. RNA-sequencing, PCR and western-blots were performed. HIF1α targets were evaluated by immunohistochemistry in bleomycin-treated rats with/without nintedanib and in patient samples with IPF. HLFs cultured on IPF-CM showed over-expression of ‘HIF1α signaling pathway’ (KEGG, *p* < 0.0001), with emphasis on *SERPINE1* (PAI-1), *VEGFA* and *TIMP1*. IPF patient samples showed high HIF1α staining, especially in established fibrous tissue. PAI-1 was overexpressed, mainly in alveolar macrophages. Nintedanib completely reduced HIF1α upregulation in the IPF-CM and rat-bleomycin models. IPF-HLFs alter the extracellular matrix, thus creating a matrix that further propagates an IPF-like phenotype in normal HLFs. This pro-fibrotic loop includes the HIF1α pathway, which can be blocked by nintedanib.

## 1. Introduction

Idiopathic pulmonary fibrosis (IPF) is a progressive and irreversible lung disease [1]. Fibrosis, in response to tissue damage or persistent inflammation, is a pathological hallmark of many chronic diseases [2,3]. Myofibroblasts accumulate in fibrotic lesions and secrete excessive extracellular matrix (ECM) proteins, resulting in distortion of pulmonary structure and scar tissue formation [4]. In addition, they recruit inflammatory cells by secreting various pro-inflammatory and pro-angiogenic cytokines, further aggravating the inflammatory response [5]. 

Mechanisms of fibrosis progression can be self-sustaining once established, without the need for ongoing inflammatory signaling. As reviewed by Herrera et al. [6], in the absence of exogenous cytokines, IPF-ECM alone can induce normal lung fibroblasts to turn into activated myofibroblasts. Once formed, IPF-ECM sets up a pro-fibrotic feedback loop that is capable of sustaining progressive fibrosis. In support of this hypothesis, our group developed an in-vitro IPF conditioned matrix (IPF-CM) system, which utilizes primary human lung fibroblasts derived from patients with IPF (IPF-HLFs). Normal HLFs (N-HLFs) that are added to the established IPF-CM, presented with increased migration and elevated pro-fibrotic markers. This platform allows us to investigate drug function with regard to fibroblast-ECM interaction [7]. 

To date, nintedanib and pirfenidone are the only approved treatments for IPF, yet they only slow disease progression. Nintedanib is known to inhibit the receptor tyrosine kinases of platelet-derived growth factor (PDGF), fibroblast growth factor (FGF), and vascular endothelial growth factor (VEGF) [8,9]. However, other non-receptor tyrosine kinases and additional targets were suggested to play a role in exerting its antifibrotic effects. For example, nintedanib was shown to mediate the inhibition of TGF-β1—induced myofibroblast differentiation via SMAD3 and MAPK inhibition [10] TGF-β1, which is the most potent inducer of fibrogenesis, inhibits PHD2 expression and induces HIF1A stabilization in fibroblasts [11], suggesting that the HIF signaling pathway can also be activated without prominent hypoxic conditions. 

The relatively high incidence of systemic side effects warrants the development of more efficient, preferably localized, treatments. For this aim, an inhaled version of nintedanib was recently described. Using two animal models [12,13], it was shown that oral-equivalent to superior effects could be obtained administering substantially reduced inhaled doses, with less adverse effects on general health. Moreover, the activity of inhaled pharmacokinetics were further supported using the in-vitro IPF-CM system wherein only short-duration nintedanib exposure was required to achieve similar effects [13]. 

In the current work, we used RNA-sequence analysis from the IPF-CM model to explore the underlying signaling pathways and characterized nintedanibs’ involvement in this process. Results were validated using IPF patient samples and bleomycin-treated animals. 

## 2. Results

### 2.1. The HIF1A Pathway Is Over-Expressed in N-HLFs Exposed to IPF-CM

N-HLFs cultured for 24 h on IPF or control N-CM (*n* = 5) were subjected to RNA-seq analysis. A total of over 20,000 genes were discovered, with 234 significantly upregulated targets. Wong et al. recently published a combined list of “Known Pulmonary Fibrosis Genes”, which consists of human genes associated with pulmonary fibrosis from several datasets [14]. We compared the 234 upregulated genes to their list and found 25.6% (60 genes) were related to pulmonary fibrosis (Figure 1A). Analysis of these 234 genes showed significant enrichment for “response to hypoxia” (GO:0001666) and ‘response to decreased oxygen levels’ (GO:0036293) (*p* < 0.0001, FDR = 0.003) using WebGestalt [15] (Figure 1B). Additionally, David [16] analysis showed the “cellular response to hypoxia” (GO:0071456) as the most significant term (*p* = 8.3 × 10^−5^). KEGG pathway analysis showed that the “HIF1α signaling pathway” was the most significant (FDR = 0.000011174, Enrichment ratio = 8.8089). The upregulated genes related to the HIF1α pathway are listed in Table 1. This list included several target genes that are implicated in both angiogenesis and vascular endothelial growth factor (VEGF) pathways (*FLT1*, *VEGFA*, *SERPINE1* and *TIMP1*). These targets were previously implicated in fibrosis [17,18,19], presented with a relatively high number of reads, and were thus validated by qPCR (*p* < 0.05, Figure 1C). Although *GAPDH* levels were elevated in the analysis, it was previously shown not to be significantly altered by western blot [7]. For validation purposes, additional house-keeping genes (*HPRT1* and beta-actin (*ACTB*)) were added, based on the sequencing results that showed no change following IPF-CM exposure (fold change (FC) 0.98, *p* = 0.95 and 1.02, *p* = 0.69, respectively). In addition, HIF1α was also significantly elevated following exposure to the IPF-CM at the mRNA and the protein levels (*p* < 0.05, Figure 1D,E). Additionally, protein levels of plasminogen activator inhibitor-1 (PAI-1, encoded by *SERPINE1*) were elevated (Figure 1E).

### 2.2. HIF1A-Related Signaling Is Upregulated in Tissue Samples from Patients with IPF

Analysis of IPF and normal tissue samples (*n* = 20) showed that IPF tissues expressed significantly higher *HIF1A, TIMP1, SERPINE1* and *VEGFA* mRNA levels (*p* < 0.05, Figure 2A). IHC staining of paraffin tissue samples from patients with IPF and normal tissue samples revealed that HIF1α is abundant in the IPF tissue. Further analysis showed that HIF1α is mostly expressed in alveolar macrophages (AM), basal bronchial epithelial cells and stromal cells within the fibrotic areas. Interestingly, HIF1α expression was significantly weaker within fibroblastic foci, even in comparison to non-fibrotic areas of the IPF sample (*p* < 0.001, Figure 2B,E). Similarly, PAI-1 was mostly found within AMs and basal bronchial epithelial cells, also showing a similar pattern of lower staining in fibroblastic foci (Figure 2C,E). TIMP1 staining was relatively weak and mostly stromal (Figure 2D,E). 

### 2.3. Nintedanib Prevents the Elevation of HIF1α-Related Targets

Previously, we showed equivalent prevention of IPF-CM effects between long term (24 h; oral pharmacokinetic mimic) and short term (60 min; inhalation pharmacokinetic mimic) nintedanib exposure, including inhibition of αSMA and COL1A [13] that were increased as a result of IPF-CM exposure [7]. Here, we tested whether N-HLFs exposed to nintedanib presented with reduced HIF1α targets. To mimic inhalation, cells were exposed to nintedanib (1–10 nM) for 60 min prior to IPF-CM challenge. Results indicated that short-duration nintedanib exposure reduced *TIMP1*, *SERPINE1* (PAI-1) and *VEGFA* mRNA levels (*p* < 0.05, Figure 3A–C). In addition, mRNA and protein levels of HIF1α were reduced (*p* < 0.05, Figure 3D,E). Further supporting inhalation as a means to deliver effective nintedanib and correlating to changes in aggregate size and cell count, these results show that short-duration exposure is as effective as continuous exposure in this human derived in-vitro IPF model. 

### 2.4. Rat Bleomycin Model Shows Elevated Levels of HIF1α within Fibrotic Tissue

These findings were tested in an *in-vivo* model, which included rats exposed to bleomycin, followed by treatment with oral or inhaled vehicle/nintedanib [12]. Results showed that nintedanib was associated with lower numbers and smaller foci of fibrosis resulting in a lower median fibrosis score for animals in this group compared to inhaled vehicle treated controls. Here, we evaluated HIF1α, TIMP1 and PAI-1 in cell blocks from those experiments. In support of the above findings, HIF1α expression was significantly elevated in rats exposed to bleomycin, whereas treatment with nintedanib significantly reduced these effects (ANOVA *p* < 0.0001, Figure 4A). TIMP1 was weakly detected in a cytoplasmic staining in all samples and was also observed in mast cells within the fibrotic areas. PAI-1 was also increased in the bleomycin treated rats, yet to a lesser extent and was slightly reduced with nintedanib treatment. Interestingly, the high inhaled nintedanib dose (0.375 mg/kg) increased PAI-1 levels (*p* < 0.05, Figure 4B), highlighting the importance of low dosing whenever possible. 

## 3. Discussion

The pathogenesis of IPF is still incompletely understood, and is thought to involve recurrent injury to the alveolar epithelium inducing an aberrant wound healing response, characterized by activation and proliferation of fibroblasts and myofibroblasts [20]. With the aim to discover underlying mechanisms leading to myofibroblast differentiation, we used our established human IPF-CM in-vitro model, which is a platform that enables the investigation of fibroblast-ECM interactions. In the high-throughput RNA analysis, the HIF1α pathway was found to be the most significant. 

HIF1α, a key transcriptional factor in response to this chronic hypoxia, is involved in many fibrotic diseases [21,22] and was shown to regulate the expression of over 200 genes in an oxygen-dependent and independent manner [23,24]. The presence of hypoxia is an established hallmark of chronic tissue injury and fibrosis [25,26]. McDonough et al. [27], analyzed microRNAs and target genes from different IPF stages. Their analysis showed 20 downregulated genes, all regulated by miR-21, which is well established in IPF [28]. Pathway enrichment for these 20 genes revealed that they were related to the HIF1α signaling pathway ([KEGG]: 04066), as found in our analysis. HIF1α was also shown as one of the major genes implicated in IPF in a Single-Cell Transcriptomic Analysis by Reyfman et al. [29]. 

HIF1α was shown to promote the fibrotic phenotype of alternatively activated macrophages through increased cell differentiation profibrotic mediator production, such as IL-6 [30]. Data from pulmonary fibrosis mouse models [31] and IPF patients have revealed increased HIF1α expression in alveolar epithelial cells at an early stage of pulmonary fibrosis [32]. However, HIF1A and HIF2A double knockout in ATII cells did not show protection against lung fibrosis induced by bleomycin in mice, suggesting that HIF signaling may not have important roles in ATII during the initial developmental steps of pulmonary fibrosis [33]. However, HIF1α may act as an amplifier of the disease in fibroblasts, as suggested in our system [34]. 

Another interesting observation from our analysis was regarding the housekeeping genes. The notion of housekeeping genes has been in use in the literature for nearly 50 years. In particular, several genes have been widely used as internal controls in experimental expression studies, such as GAPDH, tubulins and 18S [35]. In our laboratory, we successfully used the GAPDH and αTubulin, which showed no significant changes. Following the RNA-sequencing analysis, discovering that *GAPDH* is slightly elevated following IPF-CM exposure, we added *HPRT1* and *ACTB* to our data, as they showed no change. Since performing sequencing for every experiment is not optional, this point should be carefully addressed in future experiments. 

When analyzing IPF patient samples, HIF1α was expressed at lower levels in fibroblastic foci, and at higher levels in collagenous “old fibrosis” areas. Supported by other works, HIF1α has also been observed to be highly expressed in various cell types throughout the lung tissue, especially within the fibrotic areas [33]. In the rat bleomycin-treated tissue, HIF1α was also significantly elevated. However, since the UIP pattern is specific to humans [27], similar analysis of fibroblastic foci vs. fibrotic tissue couldn’t be performed. 

TGFβ and hypoxia signaling appear interrelated wherein Smad3, the major TGFβ-responsive transcription factor, can be up-regulated by hypoxia [36] and collagen expression by TGF-β1 can be decreased by inhibiting HIF1α [37]. Given these and other observations, Goodwin et al. suggested that fibroblast-specific HIF1α signaling is a critical component of pulmonary fibrosis [26]. Similar to IPF-CM stimulating elevated αSMA in N-HLFs, they and others [38] showed that hypoxia can induce αSMA expression in N-HLFs. Interestingly, recent studies have shown that HIF1α expression could be increased by additional non-hypoxic stimuli (e.g., PDGF, TNFα, angiotensin II and a variety of growth factors) [34,39]. Thus, further research is needed to understand the trigger for HIF1α overexpression. 

Nevertheless, Goodwin et al. suggested that fibroblast-specific HIF1α signaling is a critical component of pulmonary fibrosis [26]. Similar to IPF-CM stimulating elevated αSMA in N-HLFs, they and others [38] showed that hypoxia can induce αSMA expression in N-HLFs. Interestingly, recent studies showed that HIF1α expression could be increased by additional non-hypoxic stimuli (e.g., PDGF, TNFα, angiotensin II and a variety of growth factors) [34,39]. Whatever the trigger, HIF1α and its target genes (*SERPINE1, VEGFA* and *TIMP1*) were significantly overexpressed in the IPF-CM-exposed N-HLFs. As previously described by McMahon et al. [11], factors such as TGFβ1 stabilizing the HIF1α, may also be implicated in the pro-fibrotic phenotype induced by IPF-CM. Our previous work showed that the TGFβ pathway can be activated by the IPF-CM [7]. 

Preclinical and clinical studies show that the fibrotic process correlates with the expression of HIF1α target genes, such as TIMP1 and PAI-1 [32,40]. Among the numerous genes whose expression is regulated by TGFβ1 and HIF1α, PAI-1 has been shown to play a key role in pulmonary fibrosis development [39,41]. Moreover, bleomycin-induced fibrosis was shown to be more severe in transgenic mice overexpressing PAI-1, whereas PAI-1 knockout mice were protected from fibrosis. Ueno et al. showed that PAI-1 expression was highest on day 7 following bleomycin exposure and reduced thereafter [39]. Since our analysis was performed on day 28, this may explain the relatively low levels detected. Interestingly, it was also shown that TGFβ1 exposure increased PAI-1 and HIF1α protein, but not *HIF1A* mRNA levels [39]. These findings support our results, which suggest the HIF1α overexpression is probably controlled at the post-transcriptional level.

VEGFA, another HIF1α target was also elevated following HLF exposure to the IPF-CM, as well as in tissue lysates of patients with IPF. These findings support other works proposing that VEGFA might facilitate fibrogenesis [42,43]. However, as recently reviewed by Barratt et al. [19], it was stated that the relationship of VEGFA expression in IPF remains controversial and appears to differ according to the compartment sampled. Therefore, it remains unclear whether the upregulation in VEGFA is a protective mechanism or an inducer of fibrosis [44]. In the IPF-CM system, *VEGFA* mRNA expression was significantly blocked by nintedanib, which is known to inhibit the VEGF receptor [8,9]. Since VEGFA is a secreted factor, HLF protein expressions were not evaluated following culture on the IPF-CM. Additionally, since BALF or serum samples were not available, VEGFA levels were not further analyzed in the rat model. 

In the last part of this work, the in-vivo impact of both oral and inhaled nintedanib on bleomycin-induced HIF1α expression was studied. Similar to the histopathologic findings, both routes of nintedanib administration inhibited HIF1α to a similar extent, with the inhaled high dose being most effective. However, PAI-1 differed where the higher inhaled dose showed elevated levels. One explanation could be that the high dose inhibited the progression of disease, and thus the high amounts represent a slower timeline of disease. To test this theory, earlier sections (from days 7–14) are needed that might show reduced PAI-1 expression with the high doses. Alternatively, the higher dose may have initiated additional signaling pathways, such as signal transducer and activator of transcription 3 (STAT3). STAT3 was previously shown to non-canonically activate Smad3 and therefore, possibly, PAI-1. In our recent work we showed that nintedanib can elevate pSTAT3 levels in HLFs [45]. 

TIMP1 was specifically observed in mast cells within the fibrotic areas. These findings correlate to a recent study by Overed-Sayer et al. [20] that showed induction of mast cells in IPF and could explain the lack of positive staining in the HLFs. Moreover, this group showed that nintedanib inhibits mast cell survival and activation and thus provides a novel additional mechanism by which this drug may exert anti-fibrotic effects in patients with IPF.

Stabilized HIF1α under hypoxic conditions can trans-activate target genes involved in metabolic reprogramming into anaerobic glycolysis [46]. In fact, it was recently suggested that glycolytic metabolism may be the next focus point in pulmonary fibrotic progression. Zhao et al. recently showed that fibroblast catabolism of ECM could limit fibrotic progression [47]. Furthermore, there is growing literature examining the potential of targeting metabolism to treat “non-metabolic” conditions, such as fibrosis [48]. Accordingly, high lactic acid levels, indicative of up-regulated glycolysis, have been observed in patients with IPF, and were associated with augmented myofibroblast differentiation [49]. These results suggest that glycolytic reprograming is an important contributor to myofibroblast activation and differentiation, thereby promoting pulmonary fibrosis progression. Notably, the second most upregulated pathway in our RNA-sequencing analysis was “Glycolysis/Gluconeogenesis”, including the SLC2A1 (GLUT1) gene, which was previously implicated in fibrosis progression and also known to be induced by TGFβ [50]. This direction was not explored in the current work, but highlights the complexity of events leading to fibrosis. 

In conclusion, once formed, IPF-ECM sets up a pro-fibrotic feedback loop that is capable of sustaining progressive fibrosis. This loop was shown to include the HIF1α signaling pathway. To our knowledge, we are the first to show the inhibition of this pathway by nintedanib. Although the complete mechanism was not fully elucidated, nintedanib was shown to reduce HIF1α levels and to block the pro-fibrotic process propagated by the ECM. 

## 4. Materials and Methods

### 4.1. Primary HLF Culture

Primary HLFs were isolated from IPF (histologically confirmed) tissues and control samples (histologically normal lung distant from the resected tumor) at the time of biopsy, as described [51]. Following extraction, HLFs were cultured in Dulbecco’s Modified Eagle’s Medium (DMEM) supplemented with 20% FCS, L-glutamine (2 mM) with antibiotics (Biological Industries, Beit-Haemek, Israel) and maintained in 5% CO_2_ at 37 °C.

### 4.2. IPF-CM Model

Experiments were performed as previously described [7]. Briefly, IPF/N-HLFs were cultured on Matrigel (BD biosciences (by Corning), Glendale, AZ, USA). Following 48 h, cells were removed by NH_4_OH and N-HLFs were added for further culture. Nintedanib (1–100 nM, Avalyn Pharma, Seattle, WA, USA) was diluted in DMSO and added an hour prior to the addition of N-HFs to the culture. 

### 4.3. Western Blot

Western blot was performed as previously described [51]. Antibodies are listed in Appendix A. Bound antibodies were visualized using Goat peroxidase-conjugated secondary antibodies (Appendix A, doi:10.6084/m9.figshare.13475100) followed by enhanced chemiluminescence detection (Millipore, Temecula, CA, USA). LAS-3000 (Fujifilm, Japan) was used to quantify protein expressions. 

### 4.4. Real Time Quantitative PCR (qPCR)

RNA was extracted by RNeasy kit (Qiagen, Germany) and converted to cDNA using GeneAmp (Applied Biosystems, Foster City, CA, USA). Reactions were performed with SYBR Green (Applied Biosystems, Carlsbad, CA, USA). Primers are listed in Appendix A, doi:10.6084/m9.figshare.13475142). 

### 4.5. RNA-Sequencing

RNA libraries were generated using CEL-Seq protocol, and sequenced on Illumina HiSeq2500, 15/50 paired-end run. Reads were trimmed using trim galore (cutadapt version 1.10) and mapped to the Human genome (ftp://ftp.ensembl.org/pub/release-89/fasta/homo_sapiens/dna/Homo_sapiens.GRCh38.dna.primary_assembly.fa.gz) accessed on 20 May 2020, using Tophat2 version 2.1.0 (uses Bowtie2 version 2.2.6, http://www.ncbi.nlm.nih.gov/pubmed/19289445?dopt=Abstract, accessed on 20 May 2020), with up to 2 mismatches allowed per read. The minimum and maximum intron sizes were set to 70 and 500,000, respectively, and an annotation file was provided to the mapper. Only uniquely mapped reads were counted to genes, using ‘HTSeq-count’ package version 0.6.1 with ‘union’ mode (http://www.ncbi.nlm.nih.gov/pubmed/25260700?dopt=Abstract), accessed on 20 May 2020. Normalization and differential expression analyses were conducted using DESeq2 R version 1.18.1 package.

### 4.6. Rat Bleomycin Model

The study consisted of 7 treatment groups with 10 animals per group as previously described [12]. Bleomycin (1 mg/kg in 100 μL) or vehicle was administered to animals on four occasions by oropharyngeal aspiration during Week 1 to induce lung fibrosis (Days 1, 2, 3 and 6). Treatments were started on Day 8 after a one-day bleomycin recovery period. Oral nintedanib was administered twice daily (BID) by oral gavage at 60 mg/kg. Inhaled nintedanib was administered once daily (QD) by oropharyngeal aspiration at 0.25 and 0.375 mg/kg. Body weights were recorded daily. On Day 28, animals were euthanized, and terminal lung and body weights were obtained. The right lung from each animal was inflated and then immersion fixed with 10% neutral buffered formalin and embedded in paraffin. Three sections for the right caudal lobe were cut for the immunohistochemistry analysis. 

### 4.7. Immunohistochemistry (IHC)

Sections prepared from patients with IPF and from the rat lungs were deparaffinized in xylene and alcohol, rinsed in PBS and immersed in citrate buffer (pH 6). Slides were incubated overnight at 4 °C with primary antibodies (Appendix A), and developed using ImmPACT DAB (Vector laboratories, Burlingame, CA, USA) according to manufacturers’ instructions. Use of isotype-matched control eliminated non-specific staining. Expression levels were measured using Qupath [52]. Sections were counterstained with hematoxylin (Sigma, St. Louis, MO, USA). Fibroblastic foci, established fibrosis and non-fibrotic areas of the IPF lung sections were annotated by a registered senior pathologist.

### 4.8. Statistical Analysis

Analysis was done using GraphPad Prism (GraphPad Software, San Diego, CA, USA, www.graphpad.com, accessed on 20 May 2020) and by SPSS (IBM, Armonk, NY, USA). ANOVA was performed to compare differences between multiple cohorts. Student’s *t*-tests were employed to analyze differences between two groups. An effect was considered significant for *p* < 0.05. 

### 4.9. Ethics Statement

The HLF study was approved by the Ethics Committee of the Meir Medical Center (MMC-016-16, issued on 30 July 2016). Informed consent was obtained from all patients. All animal experiments were performed in accordance with the UK Animals (Scientific Procedures) Act, 1986, under the authority and controls of a project license held at the Charles River Edinburgh facility. 

## Figures and Tables

**Figure 1 ijms-22-03331-f001:**
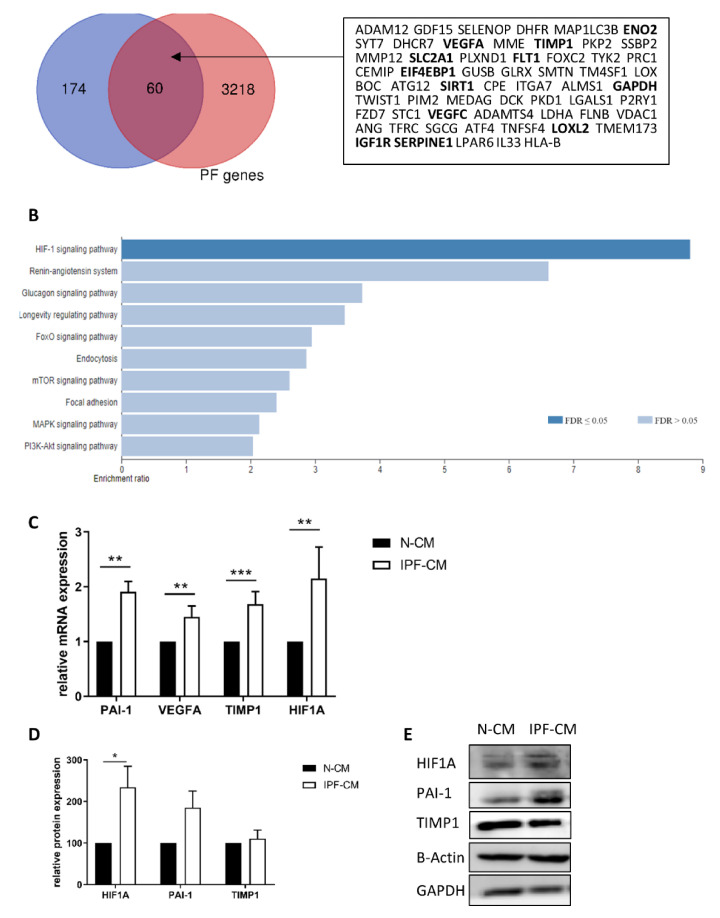
The HIF1α pathway is over-expressed in N-HLFs exposed to IPF-CM. Normal human lung fibroblasts (N-HLF) were cultured on N/IPF-CM for 24 h. Following culture, RNA was extracted for RNA-sequencing analysis. (**A**) Venn diagram of the genes elevated in our analysis vs. a list of pulmonary fibrosis (PF) genes. Shared genes are listed in the box. Genes related to the HIF1α pathway are highlighted in bold. (**B**) Pathway enrichment analysis (KEGG) using Webgestalt for the upregulated genes. *SERPINE1* (PAI-1) (*n* = 9), *VEGFA* (*n* = 11), *TIMP1* (*n* = 12) and *HIF1A* (*n* = 8) mRNA levels were measured by qPCR (**C**). The effect of the IPF-CM on N-HLF HIF1α (*n* = 12), PAI-1 (*n* = 12) and TIMP1 (*n* = 4) protein levels (24 h) were analyzed by western blotting (**D**,**E**). * *p* < 0.05, ** *p* < 0.01, and *** *p* < 0.001.

**Figure 2 ijms-22-03331-f002:**
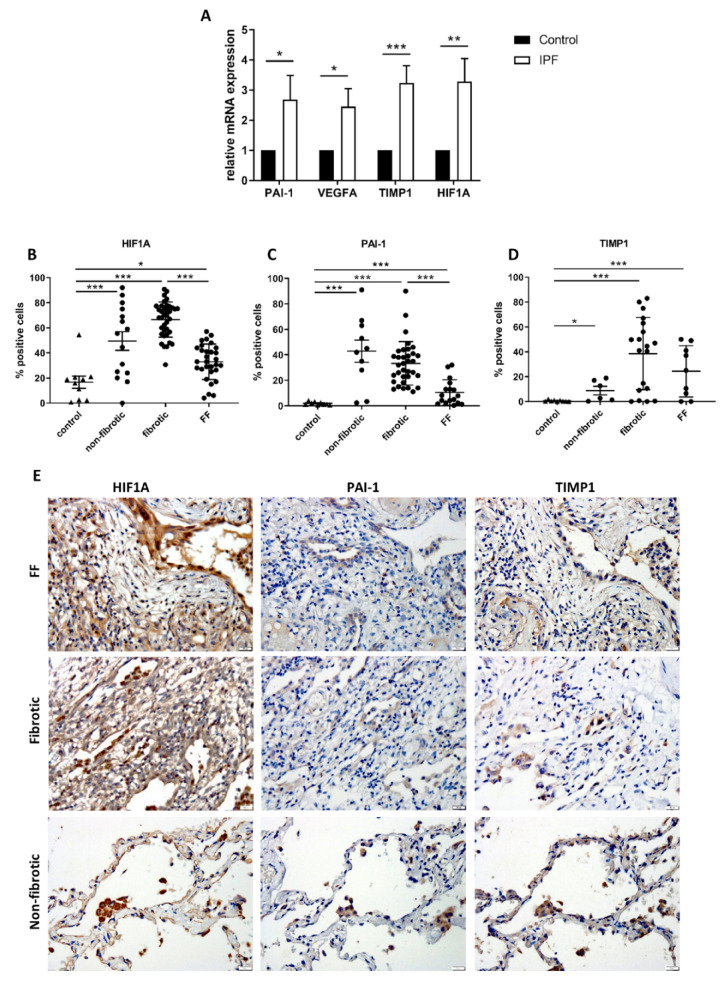
The HIF1α pathway is overexpressed in IPF patient samples. RNA was extracted from lung tissue lysates derived from patients with IPF or control donors (control). *SERPINE1* (PAI-1) (*n* = 20), *VEGFA* (*n* = 10), *TIMP1* (*n* = 30) and *HIF1A* (*n* = 30) mRNA were measured by qPCR (**A**). Formalin fixed paraffin embedded sections were stained for HIF1α (*n* ≥ 10), PAI-1 (*n* ≥ 10), TIMP1 (*n* ≥ 6) by IHC and counter stained by hematoxylin (**B**–**D**). Control tissues and Fibroblastic foci (FF), established fibrosis (fibrotic) and non-fibrotic areas of the IPF lung sections were annotated by a registered senior pathologist and quantified. (**E**) Representative IHC for figures B–D (magnification X200). * One-way ANOVA *p* < 0.0001, with the main effect of IPF versus control. Bars represent *t*-tests * *p* < 0.05, ** *p* < 0.01, and *** *p* < 0.001.

**Figure 3 ijms-22-03331-f003:**
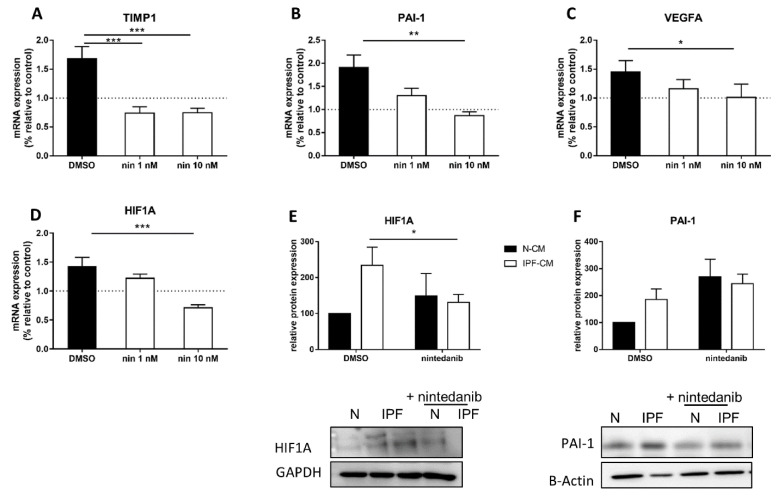
Nintedanib prevents the elevation of HIF1α-related targets. Normal human lung fibroblasts (N-HLF) were exposed to nintedanib (1–10 nM) or DMSO for 60 min and the cultured on N/IPF-CM for 24 h additional without nintedanib or DMSO. Effects of the IPF-CM with/without nintedanib (nin) on *TIMP1* (*n* = 6), *PAI-1* (*n* = 6) and *VEGFA* (*n* = 6) and *HIF1A* (*n* = 6) mRNA levels and on HIF1α (*n* = 6) and PAI-1 (*n* = 6) protein levels were tested by qPCR (**A**–**D**) and Western Blot (**E**,**F**), respectively following 24 h. Values are means ± SEM. * *p* < 0.05, ** *p* < 0.01 and *** *p* < 0.001.

**Figure 4 ijms-22-03331-f004:**
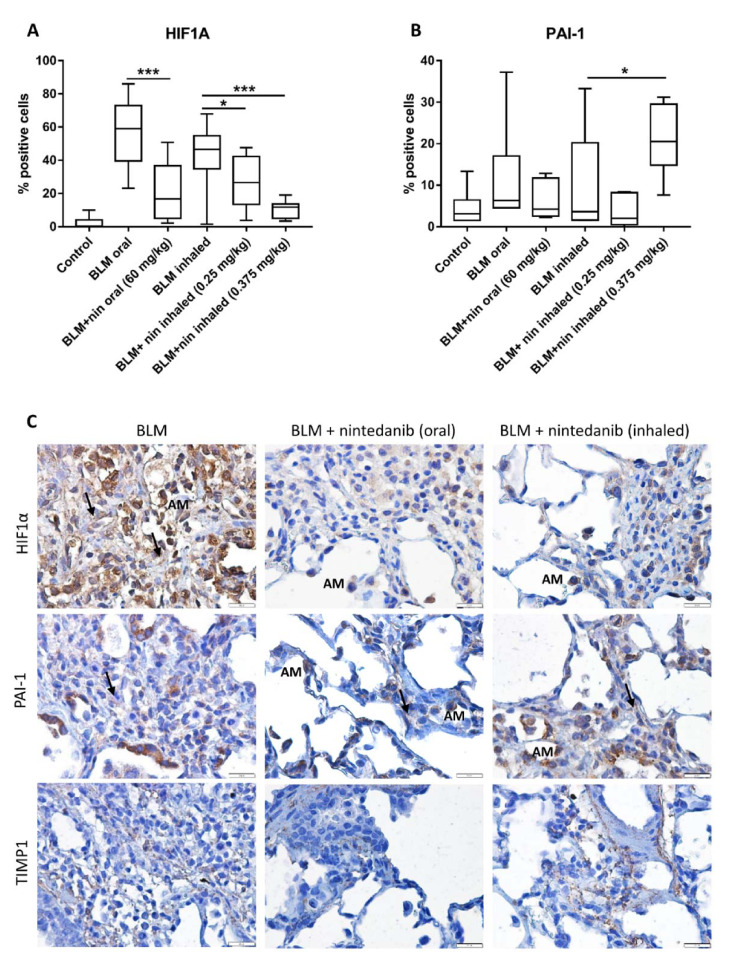
Rat Bleomycin model shows elevated levels of HIF1α within fibrotic tissue. Formalin fixed paraffin embedded (FFPE) sections of rats treated with vehicle (control), bleomycin (BLM) with/without oral or inhaled nintedanib (nin) were stained for HIF1α (**A**) (*n* ≥ 7), PAI-1 (**B**) (*n* ≥ 6) and TIMP1 by IHC and counter stained by hematoxylin. (**C**) Representative IHC for figures **A**,**B** and for TIMP1. * *p* < 0.05, *** *p* < 0.001. AM-alveolar macrophages, arrows indicate fibroblasts. Magnification (X400). Arrows indicate staining of fibroblasts.

**Table 1 ijms-22-03331-t001:** Genes upregulated in N-HLFs following culture on IPF-CM related to HIF1α signaling pathway (KEGG).

Gene Symbol	Gene Name	FC	pV	# Reads
SLC2A1	solute carrier family 2 member 1	1.456	0.00034	50–200
ENO2	Enolase 2	1.362	6.01E-06	300–800
IGF1R	Insulin-like growth factor 1 receptor	1.360	0.10716	30–80
EIF4EBP1	Eukaryotic translation initiation factor 4E-binding protein 1	1.355	0.14223	20–40
TFRC	Transferrin receptor	1.13	0.03398	570–800
VEGFA	vascular endothelial growth factor A	1.281	0.00117	1440–3400
GAPDH	glyceraldehyde-3-phosphate dehydrogenase	1.219	0.00016	9–13K
SERPINE1	serpin family E member 1 (PAI-1)	1.160	0.02021	2000–4200
TIMP1	Tissue metallopeptidase inhibitor 1	1.133	0.00372	12–25K
FLT1	Fms related tyrosine kinase 1	2.27	0.02767	3–12

## Data Availability

Any data can be supplemented on demand.

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
