# Peer review of "Hypoxia Inducible Factor 1A Supports a Pro-Fibrotic Phenotype Loop in Idiopathic Pulmonary Fibrosis"

_ijms, 2021, doi:10.3390/ijms22073331_

Round 1

Reviewer 1 Report

The article entitled, “Hypoxia inducible factor 1A supports a pro-fibrotic phenotype loop in idiopathic pulmonary fibrosis” is mainly focused on showing the effect of nintedanib (a therapeutic drug for Idiopathic pulmonary fibrosis) with inhalation as a better administration route in comparison to oral administration. The authors also presented a model known as IPF conditioned matrix to study the effect of the drugs. The study can be considered for publication after addressing following specific comments.

  1. Please remove line 36 to 44 and line 165 to 168.
  2. Figure 1: What was the reason/rationale for selecting only 4 genes for validation out of 60 PF genes upregulated in IPF-CM.
  3. Figure 1C and 1D: These can be combined in one graph.
  4. Authors used “HIF1A” and “HIF1α” interchangeably. Please use either of them to denote protein/gene.
  5. Figure 2: Authors mentioned that HIF1A expression was significantly weaker within fibroblastic foci, even in comparison to non-fibrotic areas of the IPF samples (p<0.001, Figure 2C and 2F) but the statistical bar between non-fibrotic and FF as well fibrotic and FF is not shown.
  6. Figure 2F: The scale and magnification are not shown.
  7. Line 151: Authors stated that “TIMP1 was weakly detected in all samples and was mainly observed in mast cells within the fibrotic areas.” Please indicate the mast cells with arrow in figure 4C.
  8. Figure 4C: The quality of figure is not good. Please revised the figure with better resolution, magnification and scale.
  9. Discussion: It is suggested to include an abstract diagram based on Figure 1, 2, and 3 to show the interaction of these protein in HIF1A pathway and the possible target of nintedanib.

Author Response

The article entitled, “Hypoxia inducible factor 1A supports a pro-fibrotic phenotype loop in idiopathic pulmonary fibrosis” is mainly focused on showing the effect of nintedanib (a therapeutic drug for Idiopathic pulmonary fibrosis) with inhalation as a better administration route in comparison to oral administration. The authors also presented a model known as IPF conditioned matrix to study the effect of the drugs. The study can be considered for publication after addressing following specific comments.

  1. Please remove line 36 to 44 and line 165 to 168.

Response: These were removed, sorry for that.

  1. Figure 1: What was the reason/rationale for selecting only 4 genes for validation out of 60 PF genes upregulated in IPF-CM.

Response: An explanation regarding the choice of targets was added to the results section (lines 97-100). Of the 10 genes that were overexpressed in the KEGG pathway, which are listed in table 1, we chose 4. This was based on the number of reads that were also added to Table 1. i.e. although FLT1 is very relevant, its expression is too low to detect by PCR and therefore it was not chosen for further analysis. Table 1 was moved to be near figure 1 for clarity.

  1. Figure 1C and 1D: These can be combined in one graph.

Response: This was done. Similarly, figure 2A and 2B were combined.

  1. Authors used “HIF1A” and “HIF1α” interchangeably. Please use either of them to denote protein/gene.

Response: This issue was reviewed and corrected. We also made all the gene names in italic.

  1. Figure 2: Authors mentioned that HIF1A expression was significantly weaker within fibroblastic foci, even in comparison to non-fibrotic areas of the IPF samples (p<0.001, Figure 2C and 2F) but the statistical bar between non-fibrotic and FF as well fibrotic and FF is not shown.

Response: The PV that was shown in the figures was the ANOVA analysis for all treatments together. Following this comment, we added separate indicators for each pair of treatments (t-test).

  1. Figure 2F: The scale and magnification are not shown.

Response: The scale is the little white line on the right side of each image. The magnification was added to the legend.

  1. Line 151: Authors stated that “TIMP1 was weakly detected in all samples and was mainly observed in mast cells within the fibrotic areas.” Please indicate the mast cells with arrow in figure 4C.

Response: Following the reviewers comments, new stainings were done. However, this time the mast cells were not found in the slides. Thus, an image without mast cells was added and the description was removed from the legend.

  1. Figure 4C: The quality of figure is not good. Please revised the figure with better resolution, magnification and scale.

Response: The figure was revised with new images with higher magnification and resolution.

  1. Discussion: It is suggested to include an abstract diagram based on Figure 1, 2, and 3 to show the interaction of these protein in HIF1A pathway and the possible target of nintedanib.

Response: This is an interesting idea, however during this short revision time limit this was not possible.

Reviewer 2 Report

The manuscript by Epstein Shochet et al provides data that the hypoxia-inducible factor 1A (HIF1A) is a key regulator for the progression of idiopathic pulmonary fibrosis (IPF). The authors show that Nintedanib can reduce IPF associated fibrosis in a mouse model of bleomycin-induced disease. They also show that the composition of IPF-fibroblast produced extracellular matrix (ECM) is important for HIF1A synthesis. However, the quality of the presented data is insufficient and needs to be improved.

Major criticisms:

  1. The introduction is lacking information on the available knowledge regarding the action of Nintedanib on cellular signalling, especially on the inhibition of tyrosine kinase receptors. Others have shown that Nintedanib inhibits HIF1A, VEGF and TGF-β production. None of these is mentioned in the introduction. The authors spent entire paragraph on the new form of application of the drug, which is not necessary. The authors also mentioned that the drug delays the progression of the disease, but does not provide a cure. Therefore, this reviewer does not understand why they put so much effort on this drug, rather than to extend their study on the role of IPF-ECM on fibrosis.
  2. All figures: In the figure legends, “n” is indicated as “n>4”. This is insufficient information. This is not a common practice. Legend should provide the exact number of each sub-figure.
  3. Chapter 2.1: Most of the first paragraph belongs to the Introduction or Discussion, but not to Results. It is not clear on which basis the 3 targets shown in figure 1C were selected. The Western-blots shown in figure 1D are not convincing.
  4. Figure 2: The results describe for figures 2C and 2F do not matched. Figure 2F is over-stained and the magnification is too low. In the text, the authors state that there is a significant difference for HIF1A, and PAI1, comparing fibrotic tissues to fibrotic foci. However, this is not indicated in figures 2C and 2D. It is also surprising that there is a low P-value for the comparison between non-fibrotic and fibrotic tissue for the two parameters, which is not obvious in figures 2C and 2D.
  5. Figure 3: It is interesting that Nintedanib reduces the mRNA encoding for TIMP1, PAI1, VEGFA, but the important information would come from the corresponding proteins, which are now shown.
  6. In the same figure, mRNA expression for HIF1A is missing, and the quality of the Western-blots in figures 3D and 3E are not good enough. Furthermore, there is no indication of the time at which the RNA or the protein were determined. This is important because the authors discuss that the kinetic is a critical factor for this study.
  7. Figure 4: This figure is also lacking the indication on the timing of the treatment. The histologies shown in figure 4C need to be improved. This is surprising since the same group provided much better histologies in earlier publications (Therap Adv Chronic Dis 2020). HIF1A staining is over-exposed and larger magnification would help to understand the description of these results in the text. Regarding the structural changes, in animals treated with bleomycin + Nintedanib, this reviewer does not see any beneficial effect of the drug, compared to bleomycin alone. Instead, the histologies indicate a worsening of fibrosis in animals treated with the drug.
  8. Table 1: This shows selected genes upregulated in normal fibroblasts cultured on IPF conditioned ECM. It would be important to show the effect of Nintedanib on this changes. To evaluate this data in the context of IPF, it is essential to compare what happens to these genes in IPF cells grown on ECM conditioned by normal fibroblasts. The presented data does not allow any conclusion that these changes of gene expression are IPF specific.
  9. Lines 199-205: This paragraph has to be rephrased. The arguments do not support each other.
  10. The Discussion on HIF1A is lacking a clear link to IPF, which have been published by others in patient tissues and models (e.g. Barratt et al. Respir Res 2018; Delbrel et al. Sci Rep 2018).
  11. VEGFA is known to play an important role in the regulation of micro-vascularisation under hypoxic conditions including IPF. Nintedanib is an inhibitor of VEGFA signalling. In this study, the authors show that the drug reduces VEGFA RNA, but they do not discuss this factor any further. Why did they assess VEGFA then?
  12. The results shown for PAI1 in Nintedanib treated cells is confusing. In figure 4B, the authors show that it is increasing, and in figure 4C, it seems that its expression is upregulated specifically in epithelial cells when the drug was inhaled. There is evidence from other studies that PAI1 is increasing the fibrotic process. As mentioned above, in figure 4C, it seems that treatment with Nintedanib made the tissue changes worse, rather than reducing it. May be this should be discussed in the context that the drug does not really cure IPF.

Minor criticism:

  1. Discussion: The first paragraph is the instructions from the journal in the template, which should not be included.

Author Response

The manuscript by Epstein Shochet et al provides data that the hypoxia-inducible factor 1A (HIF1A) is a key regulator for the progression of idiopathic pulmonary fibrosis (IPF). The authors show that Nintedanib can reduce IPF associated fibrosis in a mouse model of bleomycin-induced disease. They also show that the composition of IPF-fibroblast produced extracellular matrix (ECM) is important for HIF1A synthesis. However, the quality of the presented data is insufficient and needs to be improved.

Major criticisms:

  1. The introduction is lacking information on the available knowledge regarding the action of Nintedanib on cellular signalling, especially on the inhibition of tyrosine kinase receptors. Others have shown that Nintedanib inhibits HIF1A, VEGF and TGF-β production. None of these is mentioned in the introduction. The authors spent entire paragraph on the new form of application of the drug, which is not necessary. The authors also mentioned that the drug delays the progression of the disease, but does not provide a cure. Therefore, this reviewer does not understand why they put so much effort on this drug, rather than to extend their study on the role of IPF-ECM on fibrosis.

Response: A paragraph regarding Nintedanib on cellular signaling, with emphasis on TGF-β and HIF was added to the introduction section (lines 63-71).

  1. All figures: In the figure legends, “n” is indicated as “n>4”. This is insufficient information. This is not a common practice. Legend should provide the exact number of each sub-figure.

Response: This issue was corrected. N were added to each panel.

  1. Chapter 2.1: Most of the first paragraph belongs to the Introduction or Discussion, but not to Results. It is not clear on which basis the 3 targets shown in figure 1C were selected. The Western-blots shown in figure 1D are not convincing.

Response: An explanation regarding the choice of targets was added (lines 97-100). In addition, the number of reads were added to Table 1, in order to clarify the selection of targets with sufficient numbers of reads (i.e. although FLT1 is very relevant, its expression is too low to detect by PCR and therefore it was not chosen for further analysis).

Western blot representative images were replaced with new gels.

  1. Figure 2: The results describe for figures 2C and 2F do not matched. Figure 2F is over-stained and the magnification is too low. In the text, the authors state that there is a significant difference for HIF1A, and PAI1, comparing fibrotic tissues to fibrotic foci. However, this is not indicated in figures 2C and 2D. It is also surprising that there is a low P-value for the comparison between non-fibrotic and fibrotic tissue for the two parameters, which is not obvious in figures 2C and 2D.

Response: Images were replaced with new slides, with larger magnification.

The PV that was shown in the figures was the ANOVA analysis for all treatments together. Following this comment, we added separate indicators for each pair of treatments (t-test) and left the ANOVA in the legend.

  1. Figure 3: It is interesting that Nintedanib reduces the mRNA encoding for TIMP1, PAI1, VEGFA, but the important information would come from the corresponding proteins, which are now shown.

Response: PAI-1 protein levels are shown in this figure (now 3F). The protein level of TIMP1 in figure 1 was shown to not change following culture on the IPF-CM. Therefore, there was no point in inhibiting it with nintedanib in figure 3 and was thus not performed. As for VEGFA, unfortunately it was not tested by western blot, as it is probably secreted.

  1. In the same figure, mRNA expression for HIF1A is missing, and the quality of the Western-blots in figures 3D and 3E are not good enough. Furthermore, there is no indication of the time at which the RNA or the protein were determined. This is important because the authors discuss that the kinetic is a critical factor for this study.

Response: mRNA expression for HIF1A was added to the figure (now 3D). The western blot image for PAI-1 was replaced with a new gel. The time (24h) was added in the figure legend.

  1. Figure 4: This figure is also lacking the indication on the timing of the treatment. The histologies shown in figure 4C need to be improved. This is surprising since the same group provided much better histologies in earlier publications (Therap Adv Chronic Dis 2020). HIF1A staining is over-exposed and larger magnification would help to understand the description of these results in the text. Regarding the structural changes, in animals treated with bleomycin + Nintedanib, this reviewer does not see any beneficial effect of the drug, compared to bleomycin alone. Instead, the histologies indicate a worsening of fibrosis in animals treated with the drug.

Response: All histlogies were re-done. New images with larger magnification (X40) were added.

  1. Table 1: This shows selected genes upregulated in normal fibroblasts cultured on IPF conditioned ECM. It would be important to show the effect of Nintedanib on this changes. To evaluate this data in the context of IPF, it is essential to compare what happens to these genes in IPF cells grown on ECM conditioned by normal fibroblasts. The presented data does not allow any conclusion that these changes of gene expression are IPF specific.

Response: This table is drawn from the RNA-seq analysis that was performed on HLFs cultured on IPF-CM vs. HLFs cultured on N-CM. Of the 234 upregulated genes, these were the ones overexpressed in the hypoxia KEGG pathway. The PCR analysis that the reviewer commented on in comment number 3 (figure 1C-D) was a validation of these results. For this connection to be clearer, the table was moved after figure 1.

  1. Lines 199-205: This paragraph has to be rephrased. The arguments do not support each other.

Response: These lines were rephrased. In addition, more information was added, to address the following comment as well.

  1. The Discussion on HIF1A is lacking a clear link to IPF, which have been published by others in patient tissues and models (e.g. Barratt et al. Respir Res 2018; Delbrel et al. Sci Rep 2018).

Response: In addition to the first paragraph that discusses the importance of HIF in fibrosis, another section was added to emphasize the link between IPF and HIF (lines 208-216). Moreover, several new references were added, including the two that were suggested in this comments (e.g. Barratt et al. Respir Res 2018; Delbrel et al. Sci Rep 2018).

  1. VEGFA is known to play an important role in the regulation of micro-vascularisation under hypoxic conditions including IPF. Nintedanib is an inhibitor of VEGFA signalling. In this study, the authors show that the drug reduces VEGFA RNA, but they do not discuss this factor any further. Why did they assess VEGFA then?

Response: Following this important comment, we added an additional paragraph regarding VEGFA and IPF (lines 265-276).

  1. The results shown for PAI1 in Nintedanib treated cells is confusing. In figure 4B, the authors show that it is increasing, and in figure 4C, it seems that its expression is upregulated specifically in epithelial cells when the drug was inhaled. There is evidence from other studies that PAI1 is increasing the fibrotic process. As mentioned above, in figure 4C, it seems that treatment with Nintedanib made the tissue changes worse, rather than reducing it. May be this should be discussed in the context that the drug does not really cure IPF.

Response: The results of the nintedanib treatment were previously published by our colleagues (Surber MW, 2020, ref #12), which showed that nintedanib reduced the amount of fibrotic areas within the lung. We agree that the previous images were confusing, and therefore, all images were replaced in order to truly represent the state of fibrosis following BLM and nintedanib treatment.

Minor criticism:

  1. Discussion: The first paragraph is the instructions from the journal in the template, which should not be included.

Response: This part was removed.

Round 2

Reviewer 2 Report

The authors made substantial changes to the manuscript and addressed all queries sufficiently.